# Burden and risk factors of snakebite in Mopeia, Mozambique: Leveraging larger malaria trials to generate data of this neglected tropical disease

Emma O'Bryan[1]*, Saimado Imputiua[2], Eldo Elobolobo[2], Patricia Nicolas[1,2], Julia Montana[1,2], Edgar Jamisse[2], Humberto Munguambe[2], Aina Casellas[1], Paula Ruiz-Castillo[1], Regina Rabinovich[1,3], Francisco Saute[1], Charfudin Sacoor[1], Carlos Chaccour[1,4,5]

1 ISGlobal, Barcelona Institute for Global Health, Barcelona, Spain, 2 Centro de Investigação em Saúde de Manhiça, Mopeia, Mozambique, 3 Harvard T.H. Chan School of Public Health, Boston, Massachusetts, United States of America, 4 Centro de Investigación Biomédica en Red de Enfermedades Infecciosas, Madrid, Spain, 5 Clinica Universidad de Navarra, Pamplona, Spain

* e.obryan@nhs.net

**Data Availability Statement:** All data relevant to this article is publicly available here: https://dataverse.csuc.cat/dataverse/ISGlobal.

## Abstract

### Background

Snakebite is a neglected disease that disproportionally affects the rural poor. There is a dearth of evidence regarding incidence and risk factors in snakebite-endemic countries. Without this basic data, it will be impossible to achieve the target of a 50% reduction of snakebite morbidity and mortality by 2030 as set by the World Health Organization.

### Methods

This was a descriptive analysis nested in a 2021 community-based demographic survey of over 70,000 individuals conducted in Mopeia, Mozambique, in preparation for a cluster randomized trial to test an intervention for malaria. We describe the incidence rate, demographics, socioeconomic indicators and outcomes of snakebite in this population.

### Findings

We found the incidence of self-reported snakebite in Mopeia to be 393 bites per 100,000 person-years at risk, with 2% of households affected in the preceding 12 months. Whilst no fatalities were recorded, over 3,000 days of work or school days were lost with an individual household economic impact higher than that of uncomplicated malaria. 1 in 6 of those affected did not fully recover at the time of the study. We found significant relationships between age older than 15, use of firewood for household fuel, and animal possession with snakebite.

**Funding:** This work was funded and supported by Unitaid under the "BOHEMIA" grant (to RR and CCh). ISGlobal acknowledges support from the grant CEX2018-000806-S funded by MCIN/AEI/10.13039/501100011033, and support from the Generalitat de Catalunya through the CERCA Program. The Centro de Investigacao em Saude de Manhica is supported by the Government of Mozambique and the Spanish Agency for International Development Cooperation (AECID). The funders had no role in study design, data collection and analysis, decision to publish, or preparation of the manuscript.

**Competing interests:** The authors have declared that no competing interests exist.

## Conclusions

This study exposes higher than expected incidence and burden of snakebite in rural Mozambique. Whilst snakebite elimination in Mozambique seems unattainable today, it remains a preventable disease with manageable sequelae. We have shown that snakebite research is particularly easy to nest in larger studies, making this a practical and cost-effective way of estimating its incidence.

## Author summary

Snakebite is a neglected disease with a dearth of data and research funding. This study aimed to quantify the burden and identify risk factors of snakebite in a rural district in Mozambique by nesting snakebite-specific questions into a demographic survey conducted in preparation for a malaria cluster-randomized trial.

We have shown that nesting has the potential to undo the neglect of snakebite research, providing valuable results for local decision makers and aid further research. Piggy-backing onto a large well-resourced trial enabled us to reveal the burden and identify risk factors of snakebite with minimal disruption or expense to the parent trial team or participants. This method can encourage the global health community to transition into a more horizontal research approach.

We report an incidence close to 400 snakebite per 100,000 population per year in Mopeia leading to the loss of over 3,000 days of work/school. The median economic burden of snakebite per household was of US$ 17, making it almost 5-fold the cost of uncomplicated malaria cases. Although most bites occurred in those aged 20 to 25, the rate of bites per 1,000 population is much higher in those adults older than 64, this finding is also coupled with a lower recovery rate in the older age. We found no clear risk factor associated with place of bite and season.

## Introduction

Snakebite is a devastating condition that can take away lives and livelihoods, with estimated 80,000 to 138,000 deaths globally each year [1]. Yet, evidence suggests that this number may be grossly underestimated because snakebite occurs most frequently in rural settings, where prevention methods are not readily available and the first point of care are often traditional healers outside the formal health system [1,2]. The economic impact of snakebite disproportionally affects the rural poor, and its associated productivity costs perpetuate the poverty traps in these communities [2].

In 2017, the World Health Organization (WHO) recognised snakebite as a priority neglected tropical disease (NTD), and in 2020, set a target to reduce its morbidity and mortality by 50% by 2030 [3]. Progress towards this goal requires robust, reliable baseline data on snakebite burden [4]. However, there are still very few research projects focusing solely on snakebite at the moment. A potential solution for this scarcity of data is nesting snakebite research in other global health programmes, leveraging their infrastructure and optimising investments.

In Mozambique, the burden of snakebites had been previously estimated around 7,000 cases and 319 deaths annually [5], yet a recent community-based survey conducted in Cabo

Delgado, estimates the national number of cases to be 10-fold higher and the number of deaths to be closer to 9,000 a year [6].

This study took place in Mopeia in the central province of Zambezia, Mozambique, nested in a demographic survey deployed in 2021 in preparation for a large cluster randomized trial to assess the potential impact of mass drug administration of ivermectin to reduce malaria transmission. It is one of the largest community-based studies of snakebite undertaken to date. Using the infrastructure developed for the larger trial allowed for efficient data collection from a very rural area without requiring any extra funds. Furthermore, in Mopeia, snakebite had not been previously flagged as a major public health problem hence it departs from previous studies conducted in response to local concerns about snakebite that can overestimate the national incidence when extrapolated to the countrywide level [7].

The primary objective of this work was to provide local decision makers with descriptive data on the snakebite burden in Mopeia, specifically, the incidence of snakebite envenomation collected retrospectively through a community survey.

## Methods

### Ethics statement

The study protocol was approved by the Internal Scientific Committee and Institutional Review board from the Centro de Investigacao em Saude de Manhica (Ref: CIBS-CISM/004/ 2021), Hospital Clinic of Barcelona Clinical Research Ethics Committee (Ref: HCB/2019/ 0938) and The Ethics Research Committee of the WHO (Protocol ID: ERC.0003265).

### Study population, area and sampling

This study was nested in the demographic survey conducted in preparation for the Broad One Health Endectocide-based Malaria Intervention in Africa (BOHEMIA) cluster randomised clinical trial, which aims at assessing mass drug administration of ivermectin as a potential new tool for malaria control [8]. The survey collected data at the household and individual level between June and November 2021 in Mopeia, Mozambique.

Mopeia is a rural district in Zambezia province, Mozambique. It has a surface of 7,671 km$^2$ and it is naturally divided as the highlands of the north and the floodplains of the south. The population is dispersed and the central/south floodplains have much lower population density than the northern part traversed by the national road N1. The population of Mopeia was officially estimated as 153,355 in 2017 [9] and in 2021 the population was censed giving 131,818 [10]. Almost 50% of the population is under the age of 16 and over 80% of all head of households in Mopeia are subsistence farmers [11]. As in other rural areas of Mozambique, Mopeia has a high burden of malaria, HIV, tuberculosis and other communicable diseases which pose a heavy burden on the local economy [12]. **Table 1** provides basic socio-economic data at household and individual levels in Mopeia relevant for this snakebite analysis. A detailed socio-demographic description of Mopeia has been recently published by Ruiz-Castillo et al [10].

**Table 2** shows the medically significant snake species likely to be found in Mopeia, taken from data produced by Longbottom et al in 2018 [13], cross-referenced with Sprawls [14] and the WHO Snakebite information and data database [15].

Given the lack of geo-localisation and demographic data for the households of Mopeia, an enumeration of the households and the population was conducted in advance. 25,550 households and 131,818 individuals were registered. With this, 162 random clusters were created for the study, the sizes of which were determined by the population density of children under five years old living in the area (**Fig 1**). The creation of the clusters was not restricted nor stratified

**Table 1. Basic socio-economic data from Mopeia at household and individual levels.**

| Household characteristic (N=25550) | | Percentage (%) |
|---|---|---|
| Head of household with any formal education [12] | | 40.5 |
| Head of household Farmer [12] | | 82.6 |
| House type [10] | Traditional mud house | 36.9 |
| | Hut | 29.0 |
| | Precarious | 19.5 |
| | Conventional house | 13.2 |
| | Other | 1.3 |
| | Unknown | 0.1 |
| Main water source for cooking and hygiene [10] | Hole protected with hand pump outside | 50.6 |
| | Unprotected well outside | 16.6 |
| | Other | 32.7 |
| Time to water source [10] | Under 10 min | 31.8 |
| | Between 10-30 min | 45.6 |
| | Between 30-60 min | 17.7 |
| | More than 60 min | 4.8 |
| | Unknown | 0.03 |
| Main source of energy for lightning [10] | Batteries | 68.8 |
| | Electricity | 11.7 |
| | Firewood | 12.0 |
| | other | 7.6 |
| Livestock ownership [10] | No livestock | 92.1 |
| | Pigs | 7.4 |
| | Cattle | 0.3 |
| | Pigs and cattle | 0.2 |
| | Unknown | 0.02 |
| **Individual characteristics** (N=131818) | | **Percentage (%)** |
| Age group [10] | (0, 5) | 18.3 |
| | [5, <15) | 31.6 |
| | [15-64) | 47.4 |
| | ≥64 | 2.7 |
| Sex [10] | Male | 49.5 |
| | Female | 50.6 |

by location or any other criteria, hence the sample is geographically representative of the district. A census was carried out in the households within the cluster borders. The total number of inhabitants censed was 70,947, 54% of the district's population. The census collected data on demographics, health system usage, malaria prevention and burden of neglected tropical diseases, including snakebite. The nine specific snakebite questions covered occurrence in the previous 12 months, the location in which the bite occurred, the month it occurred, the outcome of the bite, the number of days of work or school lost and whether any livestock had been killed by snakebite (**Table 3**). All ages were included. The head of household provided written informed consent and answered household-level questions and all adults provided written informed consent to answer specific snakebite questions and to be involved in the research more broadly, assent was sought for those aged 12-17 and formal written consent was given by parent/guardian on behalf of children under 18.

**Table 2. Medically significant snake species likely to be found in Mopeia [13–15].**

| Species | Category* in Mopeia |
|---|---|
| *Atractaspis bibronii* Stiletto snake/mole viper | 2 |
| *Bitis arietans* Puff adder | 1 |
| *Bitis gabonica* East African Gaboon viper | 1 |
| *Dendroaspis angusticeps* Eastern green mamba | 1 |
| *Dendroaspis polylepis* Black mamba | 1 |
| *Dispholidus typus* Bloomslang | 2 |
| *Naja annulifera* Snouted cobra | 1 |
| *Naja mossambica* Mozambique spitting cobra | 1 |
| *Naja subfulva* Brown forest cobra | 2 |
| *Proatheris superciliaris* Floodplain viper | 2 |
| *Thelotornis mossambicanus* Eastern vine snake | 2 |

*Category as defined by the WHO [16]

CATEGORY 1: "Highest medical importance Definition: highly venomous snakes which are common or widespread and cause numerous snake-bites, resulting in high levels of morbidity, disability or mortality."

CATEGORY 2: "Secondary medical importance Definition: highly venomous snakes capable of causing morbidity, disability or death, but: (a) for which exact epidemiological or clinical data may be lacking; and/or (b) are less frequently implicated (owing to their activity cycles, behaviour, habitat preferences or occurrence in areas remote from large human populations)."

## Data collection

Data was collected by field workers through digital forms using Open Data Kit (ODK, https://opendatakit.org) in Android tablets. It was available in Portuguese and English.

To reduce the impact of recall bias, questions were only asked about events of snakebite in the previous 12 months. For those affected, questions regarding frequency, time, location, and outcome were asked. If an individual had suffered more than one episode of snakebite, they were asked to describe the most severe of the bites. When using months and location of bite for multivariable analysis, only the most recent bite was considered.

## Data management and analysis

Data collected in the field was encrypted and transferred to the local server. It was synchronised with the local and study database daily. Once complete and clean, data was uploaded into Stata version 17 (StataCorp. 2021. Stata Statistical Software: Release 17. College Station, TX: StataCorp LLC. URL https://www.stata.com/). Descriptive analysis was done via frequencies, percentages, medians and interquartile ranges. Incidence rate was calculated in person-years at risk. Logistic regressions were modelled correcting for confounders, when necessary, with odds ratios calculated with a 95% confidence interval.

## Envenomation definition

Snakebite is a bite by any snake. Envenomation is the development of local or systemic signs and symptoms. Snakebite without envenoming or "dry bite" is the absence of signs or

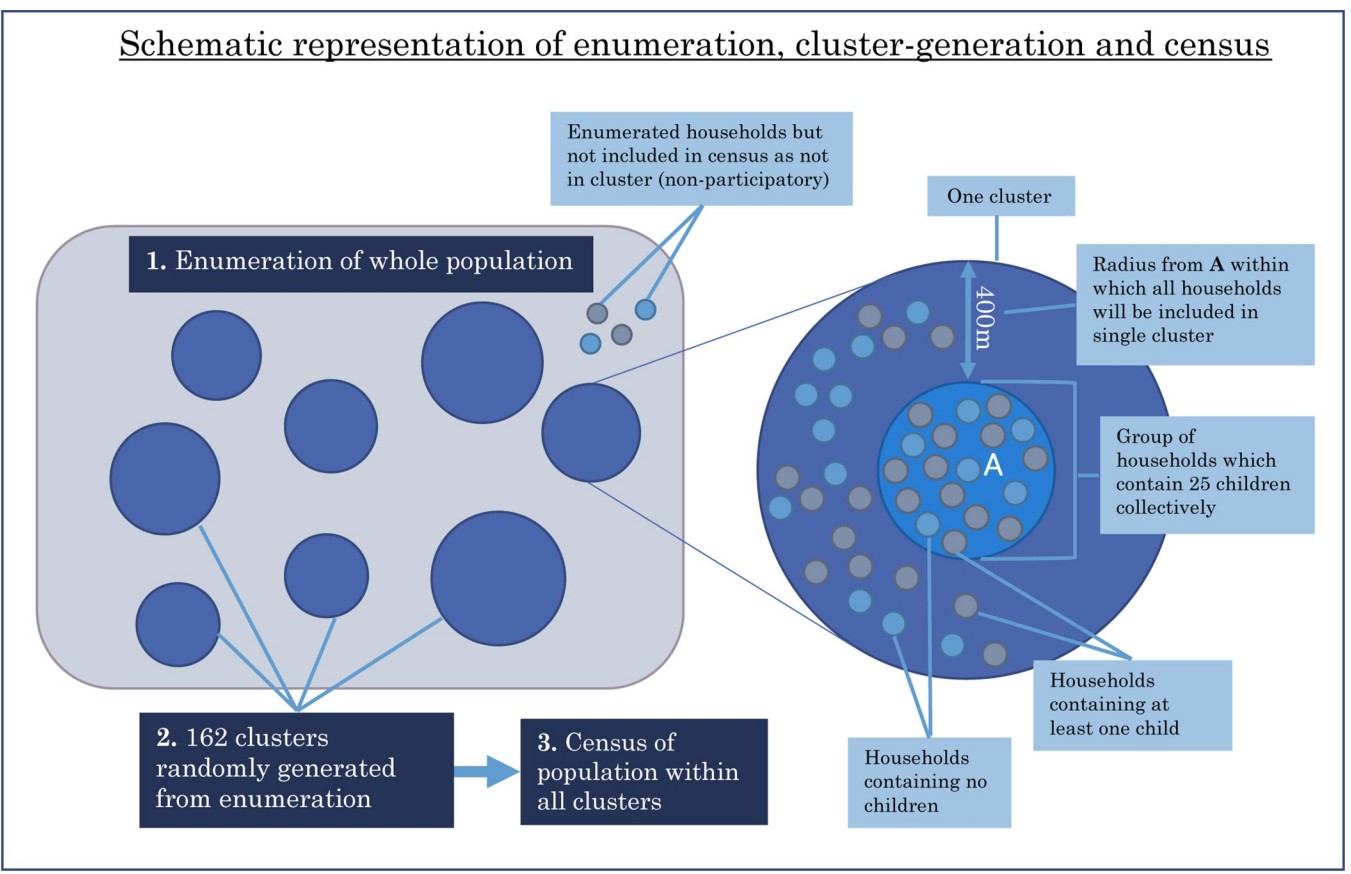

**Fig 1. Data collection sequence and cluster structure.**

**Table 3. The nine questions about snakebite embedded in the larger demography questionnaire.** Beyond these, one question about livestock morbidity/mortality also allowed for the answer "killed by a snake" but there were zero answers with that option. "Loss of limb" was operationally defined as amputation (medical or necrosis) or loss of function.

| | |
|---|---|
| 58. Has any household member been bitten by a snake in the past 12 months? | □ Yes □ No □ Don't know □ Prefer not to answer |
| 58a. [If yes] How many household members were bitten in the past 12 months? | *Integer* |
| *[Per household member bitten]: Q59a-f* | |
| 59a. Who was bitten? | *Select from list of household members* |
| 59b. How many times (separate instances) was he/she bitten? | *Integer* |
| 59c. In what month(s) was he/she bitten? (check all that apply) | □ January □ February □ March □ April □ May □ June □ July □ August □ September □ October □ November □ December □ Don't know |
| 59d. Did this person miss school/work days because of the bite? (If bitten more than once, answer about the most severe bite) | □ Yes □ No □ Don't know |
| 59d(i). [If yes] How many days? | *Integer* □ Don't know |
| 59e. Where did the bite(s) occur? (check all that apply) | □ Inside the home □ Inside the compound □ Field □ Road □ Other (specify) □ Don't know |
| 59f. What was the outcome of the bite? (If bitten more than once, answer about the most severe bite) | □ Death □ Loss of limb □ Full recovery □ Partial recovery |

symptoms in presence of fang marks [17]. Given the short length of the questionnaire, for the purpose of analysis, we used missing time from school/work or incomplete recovery at the time of the survey as proxies for systemic signs/symptoms and participants with these findings were accounted for as envenomation while those not reporting missing school/work or any sequelae at the time of questioning were accounted for as dry bites.

## Maps and geo-location

Households' longitude and latitude were collected by fieldworkers using GPS-enabled tablets, validated through automated maps and manually inspected by fieldworkers and data managers. Maps were created and stylized using RStudio (RStudio Team 2022. RStudio: Integrated Development Environment for R. RStudio, PBC, Boston, MA URL. http://www.rstudio.com/); the shapefile was obtained from the GADM database, URL https://gadm.org/data.html; the line data was obtained from OpenStreetMap, URL https://www.openstreetmap.org/.

# Results

## General sociodemographic data

A total of 70,947 individuals and 13,140 households were included. The median age of the population was 15.6 years (IQR 7.3 -28.0). Just under half of the population (48.2%) were under 15 years old, 48.9% were of working age (15-64 years) and 50.9% were female. Regarding latrines, 57% of all households did not possess any of which 86.1% practised open defecation. As a proxy for wealth, 29.7% of households possessed none of 14 pre-defined commodities (bicycle, cell-phone, vending stall for business, motorcycle, car, truck, animal-drawn cart, boat with motor, radio, television, video/DVD player, fridge, freezer and bank account) and 11.6% did not possess a bed net. Extensive details on Mopeia's socio-economic structure can be found in Ruiz-Castillo et al [10].

## Individual analysis

**General snakebite data.** A total of 272 individuals from 254 different households reported to have suffered snakebite in the previous 12 months. Of these, 5 individuals were bitten twice and one individual was bitten three times bringing the total number of bites that occurred to 279. With the denominator as the study population (70,947) this gives an incidence of 393 bites per 100,000 person-years at risk.

Using missed school/work days and incomplete recovery at the time of the study to define envenomation, 210 (77%) of the bites resulted in envenomation. The resulting incidence is 296 envenomations per 100,000 persons per year. All those who were bitten survived, 17.3% reported not making a full recovery at the time of survey.

**Demographics of those bitten.** Just under half of those bitten (132, 48%) were female, there was no significant relationship between sex and the odds of being bitten. The median age of women bitten was 29.1 years (IQR 20.2.-43.1) and of men was 29.3 (IQR 18.5-46.29). The rate of snakebite by 1000 person-years at risk significantly increased with age (**Table 4**). There were only 3 bites in children younger than five years of age and 46 in children 5-15 years old. The bulk of bites occurred in adults of work age and the most bites were suffered by those aged 20-25 (30 affected, 14 male and 16 female). A histogram with the distribution by age of all snakebite victims is presented in **Fig 2**.

**Consequences of snakebite.** The majority (225; 83%) of those bitten made a full recovery, 4 individuals (1.5%) lost a limb (**Table 5**). The rate of full recovery by age is presented in

**Table 4. Sex and age of all snakebite victims and all participants.**

| | Individuals effected by snakebite n (%) N=272 | All participants n (%) N=70,947 unless stated | Rate per 1000 person-years | Crude OR (95% CI) | P-value |
|---|---|---|---|---|---|
| **Sex** | | | | | |
| **Male** | 142 (52.2) | 34,872 (49.1) | 4.07 | Reference | |
| **Female** | 130 (47.8) | 36,075 (50.9) | 3.60 | 0.88 (0.70-1.12) | 0.31 |
| **Age cohorts** | | | | | |
| **<5** | 3 (1.1) | 11782 (16.6) | 0.25 | 0.12 (0.03-0.33) | <0.0001 |
| **5-<15** | 46 (16.9) | 22,398 (31.6) | 2.05 | Reference | - |
| **15-64** | 204 (75.0) | 34,663 (48.9) | 5.88 | 2.88 (2.11-4.01) | <0.0001 |
| **>64** | 19 (7.0) | 2,104 (3.0) | 9.03 | 4.43 (2.53-7.45) | <0.0001 |

**Table 5**. There was a statistically significant higher proportion of victims with incomplete recovery in older ages).

Almost 75% (203) of all bitten individuals missed work or school. Of the 203 who reported missing work or school, only 168 provided an estimate of missed days. In these 168 there was a median of 7 days missed (IQR 5-15). The total collective number of reported missed days was 3,039.

**Place of biting and bed nets.** The most common (42.7%) place for snakebite to occur was in the field. Although bites occurring in field contributed the most to those with only a partial recovery, those bitten at home were significantly less likely to make a full recovery (OR 0.35; 95%CI, 0.15–0.86) and two of the four reported loss of limb occurred after bites in the home. There was no significant relationship between other locations and recovery, nor with location and season.

Regarding the question "did you sleep under a bed net last night?", 75% of the censed population responded positively. The proportion answering yes, among those affected by snakebite was 82%, this difference was statistically significant. All but one of those bitten at home reported using a bed net on the previous night.

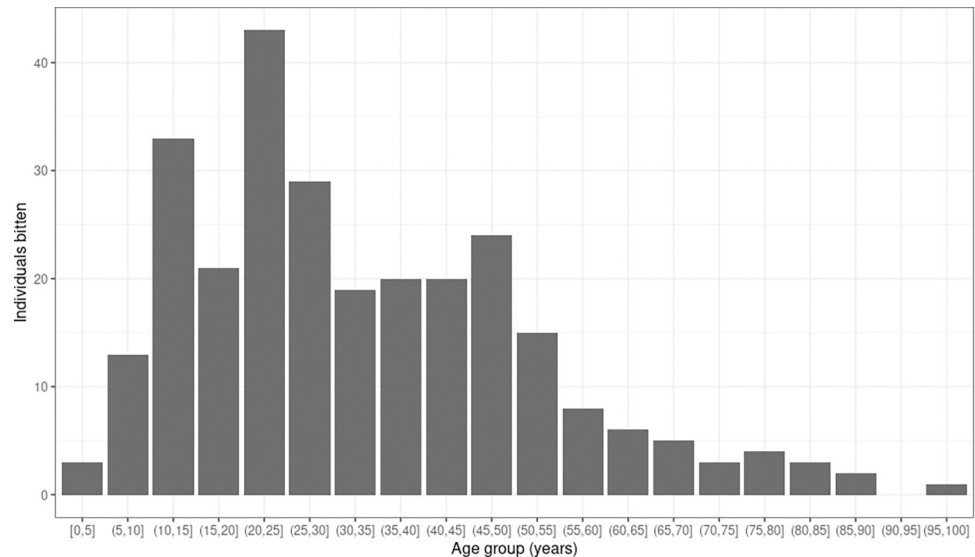

**Fig 2. Age distribution of all snakebite victims.**

**Table 5. Self-reported recovery after snakebite by age group.**

| Age (years) | Full recovery | Total bitten | Proportion recovered | OR | p-value (Chi²) |
|---|---|---|---|---|---|
| <5 | 3 | 3 | 100% | 1.02 | 0.4 |
| 5-15 | 45 | 46 | 98% | Reference | Reference |
| 15-64 | 165 | 204 | 81% | 0.83 | < 0.005 |
| >64 | 12 | 19 | 63% | 0.65 | < 0.0005 |

**Location and seasonality.** The most common month to be bitten was September and the least were July and December (**Table 6 and Fig 3**). Bites were fairly distributed along the year with 52% in the dry season, 42% in the rainy season and 6% of participants did not provide the month in which it happened. (**Table 6 and Fig 3**).

**Fig 4** shows the location of the household of those who were bitten and which month they were bitten in. Note that bites are frequently reported in households along the roads, this reflects the distribution of the population in Mopeia which is mostly clustered along primary and secondary roads. Heatmaps were constructed with these data (**Fig 5**), these reflect a higher occurrence of snakebite in the most densely populated region, Mopeia Sede, which concentrates over a third of the district´s population.

There were no differences in the number of school/work days lost due to bites in the rainy season (total = 1,412, mean = 19.08 days) versus the dry season (total = 1,598, mean = 17.75) (t-tests, p = 0.8).

## Household analysis

From the 13,140 households included in the study, there is snakebite data available from 13,119. Of these, 254 households (1.9%) suffered at least one episode of snakebite in the

**Table 6. Descriptive of consequences of snakebite, location and seasonality.**

| Consequences | n/272, (%) |
|---|---|
| Full recovery | 225 (82.7) |
| Partial recovery (excluding limb loss) | 43 (15.8) |
| Limb loss | 4 (1.5) |
| Death | 0 (0.0) |
| Productivity loss | |
| Missed work or school | 203 (74.6) |
| Median days missed of those who missed any days | 7 (5-15) |
| Collective number of work or school days missed | 3,039 |
| **Location bite occurred** | |
| Inside the household compound | 52 (19.0) |
| Field | 116 (42.7) |
| Inside the home | 25 (9.2) |
| River | 8 (2.9) |
| Road | 71 (26.1) |
| **Seasonality** | |
| Dry (May-October) | 142 (52.2) |
| Rainy (November-April) | 115 (42.3) |
| Does not remember the month | 15 (5.5) |
| Commonest month | September |
| Least common months | July and December |

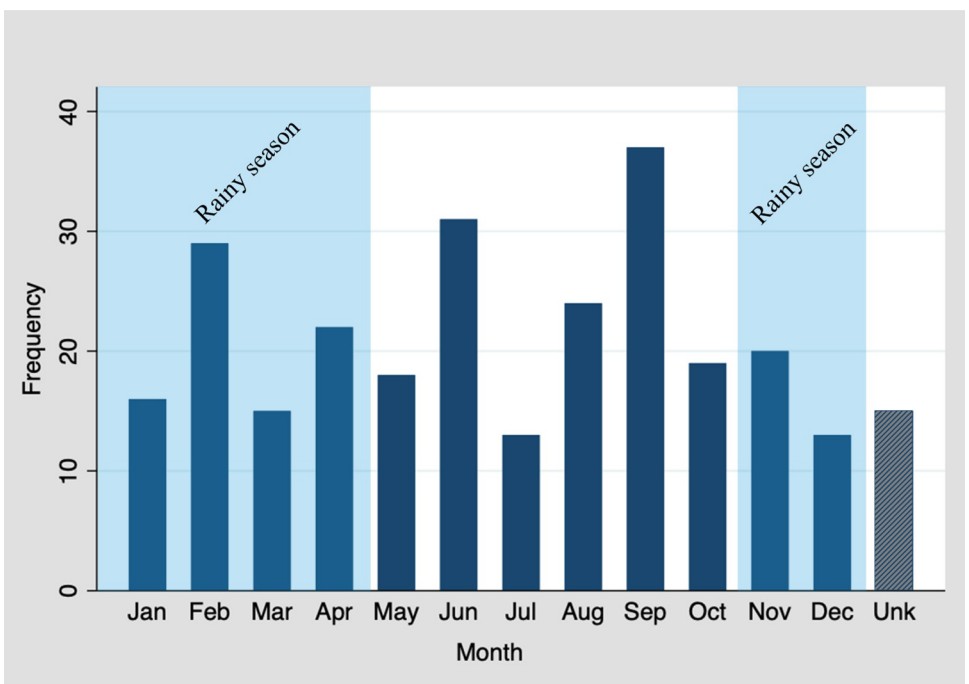

**Fig 3. Snakebite frequency by month of the year.** In Mopeia, in 2022, rains reached a peak in March with 500 mm; May was abnormally wet with precipitations of 150 mm, there were no more than 30 mm of rain per month until December.

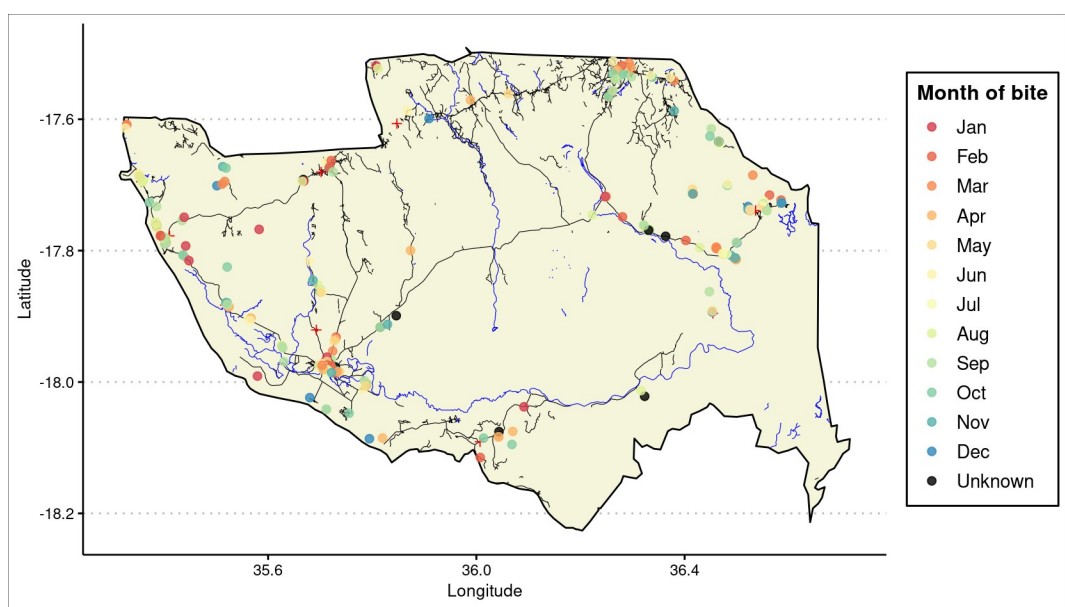

**Fig 4. Map of Mopeia showing location of households of those bitten by a snake (circles), further divided by month of bite (see key).** Black lines are roads, blue lines are rivers, red cross are health facilities. Contains information from OpenStreetMap and OpenStreetMap Foundation, which is made available under the open database license. URL https://www.openstreetmap.org/.

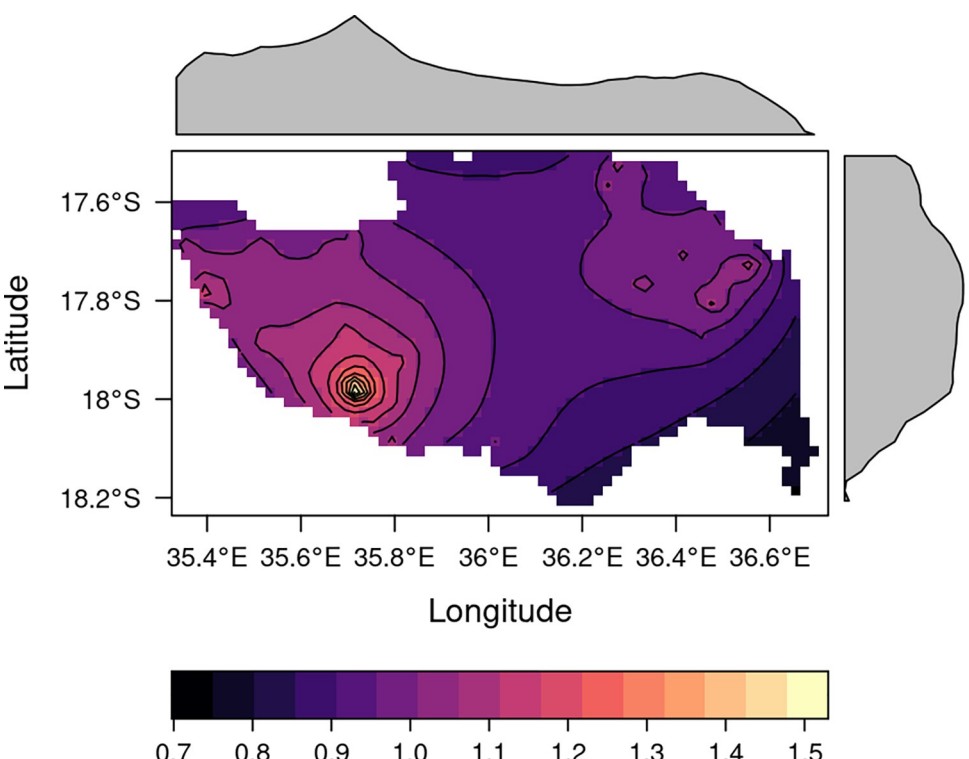

**Fig 5. Heatmap of snakebite occurrence in Mopeia.** Contains information from OpenStreetMap and OpenStreetMap Foundation, which is made available under the open database license. URL https://www.openstreetmap.org/.

preceding 12 months. One individual was affected in 245 households, eight households had two individuals affected and one household had four. **Table 7** describes the characteristics of households that have had at least one episode of snakebite compared to households with none. The odds ratios calculated are the odds of a household having at least one episode of snakebite versus none at all.

There was no significant relationship between snakebite occurrence and the composition of the household's ceiling or floor, or between snakebite occurrence and toilet practices. Possession or lack of commodities such as a bicycle or mobile phone was not significantly associated with snakebite, nor was household possession of bed nets (**Table 7**). The majority (91.4%) of all households cooked, at least in part, outdoors. The usage of firewood as fuel was found to be significant risk factor for snakebite. Household possession of any livestock and/or companion animals was found to be a risk factor for being affected by snakebite. Particularly possession of cats, dogs and goats were found to be significant risk factors for snakebite when adjusted for ownership of other animals (**Table 8**).

## Discussion

### Burden

We found an incidence of snakebite of 393 bites per 100,000 person-years at risk, largely in keeping with other community-based studies,[18–21] as well as with previous estimates for Sub-Saharan Africa[22] and Mozambique.[5,6] In Mopeia, snake bites are aligned with the distribution of the population which is clustered along the roads and more bites are reported in the district capital, where over one third of the population is concentrated.

**Table 7. Household risk factors for snakebite.**

| Characteristic | Households affected by snakebite, n/N (%) N = 254 unless stated | Households not affected by snakebite, n/N (%) N = 12,865 unless stated | Crude OR* (95% CI) | P-value |
|---|---|---|---|---|
| House materials | | | | |
| Ceiling of grass | 192 (75.6) | 9,192 (71.5) | 1.24 (0.93-1.65) | 0.15 |
| Ceiling of zinc | 62 (24.4) | 3,172 (24.7) | 0.99 (0.74-1.32) | 0.93 |
| Floor of sand | 107 (42.1) | 6,040 (47.0) | 0.82 (0.64-1.06) | 0.13 |
| Floor of adobe | 86 (33.9) | 3,879 (30.2) | 1.19 (0.91-1.54) | 0.20 |
| Latrine | | | | |
| Possession of latrine | 112 (44.1) | 5,523 (42.9) | 1.05 (0.82-1.37) | 0.71 |
| Without latrine who practise open defecation | 127/142 (89.4) | 6.313/7,342 (86.0) | 1.38 (0.80-2.37) | 0.24 |
| Cooking | | | | |
| Practise outdoor cooking | 223/253 (88.1)† | 11,773/12,849 (91.6)† | 0.68 (0.46-1.00) | **0.050** |
| Fuel for cooking | | | | |
| Firewood | 226/253 (89.3) | 10,886/12,849 (84.7) | 1.51 (1.01-1.26) | **0.045** |
| Item possession | | | | |
| No household commodities †† | 62 (24.4) | 3,836 (29.8) | 0.76 (0.57-1.01) | 0.063 |
| Possession of at least one bed net | 224/248 (90.3) ††† | 11,172/12,646 (88.3) ††† | 1.323 (0.81-1.88) | 0.34 |

* The odds ratios calculated are the odds of a household having at least one episode of snakebite versus none at all

† 17 households did not cook for themselves and the location of where the food they did eat was therefore not recorded

†† bicycle, cell-phone, vending stall for business, motorcycle, car, truck, animal-drawn cart, boat with motor, radio, television, video/DVD player, fridge, freezer and bank account.

††† 225 households did not report how many bed nets they had

A surprising result from this study is the lack of reported mortality. Mortality figures are very variable in the literature and likely influenced by the local species of snakes as well as demographic socio-economic factors. The WHO global estimates of snakebite give a case fatality rate of snakebite to be 1.5-3.1%.[1] As mortality from snakebite is highly dependent on the species, ecological surveillance of Mopeia would be needed to better understand the reasons behind the lack of mortality reported here (see below). Additionally, it would be valuable to

**Table 8. Household animal possession as a risk factor.**

| Characteristic | Households affected by snakebite, n/N (%) N = 248* | Households not affected by snakebite, n/N (%) N = 12,646* | Crude OR (95% CI) | P-value | Model** Adjusted OR (95% CI) | Model** P-value |
|---|---|---|---|---|---|---|
| Animal possession | 181 (73.0) | 7,872 (62.3) | 1.64 (1.24-2.17) | **0.001** | | |
| Animal possessed | | | | | | |
| Cat(s) | 67 (27.0) | 2,062 (16.3) | 1.90 (1.43-2.52) | **<0.001** | 1. 61 (1.19-2.18) | **0.002** |
| Dog(s) | 42 (17.0) | 1,223 (9.7) | 1.90 (1.36-2.67) | **<0.001** | 1.48 (1.03-2.12) | **0.034** |
| Goat(s) | 24 (9.7) | 665 (5.3) | 1.93 (1.26-2.96) | **0.003** | 1.59 (1.02-2.47) | **0.041** |
| Poultry | 156 (62.9) | 6,943 (54.9) | 1.39 (1.07-1.81) | **0.013** | 1.18 (0.90-1.55) | 0.24 |
| Cattle | 1 (0.4) | 28 (0.2) | 1.82 (0.25-13.5) | 0.555 | 1.11 (0.15-8.43) | 0.92 |
| Pig(s) | 23 (9.3) | 853 (6.8) | 1.41 (0.92 2.18) | 0.119 | 1.10 (0.70-1.73) | 0.67 |

*6 (0.2%) households affected by snakebite and 219 (0.2%) unaffected households did not report what animals they possessed, if any

**Model: confounders of each animal possessed, as household is likely to own one animal if it owns another

conduct a review of the local hospital records taking in consideration local beliefs and practices.

Of the venomous snake species likely to be present in Mopeia (Table 2), the WHO regards *Bitis arietans*, *Dendroaspis angusticeps*, *Dendroaspis polylepis* and *Naja mossambica* to be most important in southern Africa.[23] Of these, *D. polylepis* and *D. angusticeps* bite deliver potent neurotoxic venom often resulting in rapid death[14,23], as such we hypothesise that whilst their range includes Mopeia, it is unlikely that many if any of the bites in this study are from these species. The higher morbidity seen in those bitten at home in this study is compatible with the presence of *N. mossambica*, an aggressive species known to enter houses and whose bites often result in severe injury but not usually rapid death.[6,14,23] The highly prevalent *B. arietans* accounts for a large proportion of snakebite morbidity across the world, causing severe injury but infrequent rapid death[14,23], and is almost certainly contributing to the burden in Mopeia (in fact, the BOHEMIA study team encountered – without harm! - a *B.arietans* during this data collection). In addition to these 4 important species, we hypothesise that in Mopeia and in neighbouring areas along the Zambezi River, *Proatheris superciliaris*, whilst being a category 2 species, could be responsible for a large proportion of the bites documented in our study. *P. supercillaris* has a very limited range (only found in pockets around Lake Malawi and Lake Chilwa and on the floodplains of the Shore and Zambezi Rivers)[13,14] meaning it appears infrequently in literature. There are no documented fatalities but it can cause severe symptoms,[14] in keeping with our high burden but zero fatalities. Similarly, *Atractaspis bibronii*, found in Mopeia and across Africa, delivers severe but not fatal bites,[14] and was recently found to be the commonest cause of snakebite along with *B. arietans* in Cabo Delgado in northern Mozambique.[6] As such, we consider *N. mossambica*, *B. arietans*, *P. supercillaris* and *A. bibronii* to likely be the snakes of most concern in Mopeia.

Despite the lack of mortality, snakebite still incurs high rates of absenteeism from school and work and long-term morbidity in Mopeia. 2,643 of the total 3,039 (87.0%) days of school or work lost due to snakebite were from individuals over the age of 15. Using the Mozambique minimum wage for the agricultural sector of 2.52 USD per day,[24] the median indirect cost of snakebite due to labour losses is 17.64 USD (IQR 0-20.16 USD) per individual affected. Given that many in this area are living on under one US dollar a day, a 17 USD loss could have dramatic impact on household income. To put this in context, this cost is notably higher than the household cost associated with an uncomplicated malaria case in Mopeia (3.46 USD (IQR 0.07–22.41 USD)), but lower than the cost of a severe malaria case (81.08 USD (IQR 39.34–88.38 USD)).[25] When compared internationally, a recent study in Nepal found a lower rate of absenteeism from snakebite (23.3% vs 74.6% in this study) but the median number of days of work missed for those who missed work was the same.[26]

## Risk factors

The literature describes the typical snakebite victim in their late twenties or early thirties, either male or female, and most likely bitten in the field or bush, which is highly aligned with our own findings.[6,27,28] We, however, describe that the rate of bite per 1000 population in Mopeia is almost twice in those older than 64 years of age than the rate in those 15-64. This is an important finding given the lower rate of full recovery seen in older populations in our study. The reason for the higher rate in older individuals is worth exploring. We hypothesise it could be due to different attitudes or behaviours towards snakes in older generations or difficulty in seeing or moving away from a snake in frailer individuals.

Other risk factors for snakebite are often behavioural. It is generally agreed that activities that increase time spent outdoors increase the risk of snakebite.[2,14,28] Examples include

practicing open defecation, cooking outdoors, collecting firewood and leaving the home to fetch water. We found households who used firewood for fuel were at significantly increased risk of snakebite, possibly in association with more time spent in the bush. This adds to the many reasons why transition from firewood to gas/electric cooking is beneficial for health and development. Leaving the homes to use the toilet and practicing open defecation was not found to be a risk factor for snakebite but household sanitation facilities are fundamental to improving health, and therefore, the insignificance of these factors in this study has no practical implications. Only a small proportion (less than 10%) of the bites in our study occurred inside the home, yet these led to higher morbidity.

The general recommendation is to move away from grass and thatch roofs as a method to prevent snakebite.[29] We found no association with ceiling material and risk of snakebite. However, household modifications as a protection measure against mosquitoes are becoming more frequent, and these should also prevent the entry of larger animals such as snakes. The bed net usage among those bitten at home was higher than in the general population; this contradicts previous findings in Nepal about the protective effect of bed nets, however, this is to be interpreted carefully given the small sample size of those affected at home and the differences between Nepalese and Mozambiquan snake species. Other peri-domestic anti-mosquito measures such as cutting back long grass is also recommended in snakebite prevention[14,29]. Incorporating snakebite surveillance into home modification studies for malaria could provide valuable insight of the effectiveness of malaria interventions against snakebite and make malaria research more horizontal.

**Animal possession.**   We have found that ownership of cats, dogs and goats at the time of the survey significantly increased the risk of a household being affected by snakebite. Animal food and waste is known to attract rodents which are a common prey for snakes [14,29]. Furthermore, cats, dogs and goats are more likely to roam in and around the home, this may attract snakes into closer environments with humans. Pigs, cattle and poultry tend to be enclosed rather than roaming, therefore crossover with humans is less which could explain the lack of relationship with snakebite found here. This association of animal possession could be reverse causation as the survey asked about current animal possession and past snakebite. It could be possible that those who suffered snakebite then acquired animals as they felt it may protect the household from further snakebite, but we think this is unlikely as animal ownership has been previously found to be a risk factor for snakebite [2,14,20]. However, the association of companion animals has not been thoroughly studied.

**Geography and seasonality.**   The month in which most of the snakebites occurred was September, which is in the middle of the dry season, and the months with less snakebites reported were July and December, which represent the early dry season and the beginning of the rains, respectively. No particular seasonality of the risk or severity of snakebite in Mopeia can be inferred from our data, as seen in Northern Mozambique and Nepal [6,28] and in contrast with Ghana and Kenya where bites are more common in the wet season [20,27].

Fig 4 maps the coordinates of the households of those affected by snakebite, but not where the bite occurred. However, we can assume that individuals spend most of their time close to their homes and thus, the location of bite correlates fairly with household location. Whilst we could not find a statistical relationship between season and snakebite frequency or bite location, the visual inspection of the map leaves open the hypothesis of whether living closer to rivers can increase bite risk during the rainy season.

## Strengths

**Nesting of snakebite studies.**   Of all NTDs, snakebite is probably the easiest to understand by the communities it affects. Most communities will have a word for snakes and know to be

cautious of them. Furthermore, it is difficult to forget if you or someone in your family was bitten by a snake, so we expect limited recall bias. As such, snakebite is the ideal condition to be nested in other programmes where a community-based survey is being conducted. Unlike other NTDs, snakebite's simple definition removes the need of explanation of the disease, reducing the inconvenience and cost to both staff and participants. Whilst this study is limited by the small number of snakebite-related questions asked, we have gained useful information. We strongly advocate for nesting in research as it facilitates the move away from vertical interventions to more horizontal research practices. As such, we encourage researchers to consider nesting snakebite studies in their research where possible.

**Potential Bias.** More than half of the population of Mopeia were included in this study via randomly created clusters which reduced selection bias and gives results that are likely to be representative of this community.

The questions in this study were straightforward. Temporal and spatial occurrence of bite, recovery status and impact of professional or economic activities are simple things to remember so we anticipate low levels of recall bias due to this.

## Limitations

**Envenomation versus dry bite.** Questions relating to recovery status and missing time off work or school are not optimal to differentiate between dry bite and envenomation. The gold standard is in a clinical setting with access to laboratory investigations and snakebite experts, which is not possible in a community-based study. Here, questions regarding symptoms post-bite are better to differentiate between envenomation and dry bite. These questions were not asked in this study because it is nested in an overarching malaria trial and questions had to be rationed. As we used proxies for severity to equate envenomation, our estimations may not be fully accurate. Furthermore, no data on suspected species of snake, first aid practices or the of treatment received was collected. Going forward we recommend making every effort to include well-worded questions regarding post-bite symptoms to identify envenomation. It is useful to identify rates of envenomation within snakebite cases not only because of the more severe clinical syndrome associated with it, but also, from a public health perspective, snakebite envenomation is of particular concern due to the inequity in antivenom production and supply and in access to specialist medical treatment [2].

**Further demographic detail.** Despite the high proportion of subsistence farmers in Mopeia, no questions were asked regarding the specific level of education and profession of the persons affected by snakebite in this study. These questions have particular value in snakebite as it can be considered an occupational disease, typically affecting agricultural workers [1,2]. Similarly, whilst we have asked where the bites occurred, we have not asked the time of day nor what individuals were doing at the time of the bite. These details have been useful in determining risk factors in other studies.[18,20,28] It can be argued that as this has been found in multiple other studies their addition here would not have added to the discussion. However, we would like to have gathered data regarding perceptions of snakes and snakebite in this community as well as knowledge of snakebite first aid and what treatment the bitten individuals received in Mopeia. These questions were not included as there was not space for them in the nesting.

**Generalisability.** The results of the sub-analyses and adjusted models should be interpreted carefully given they are based on 272 individual bites. The data of this study was taken from a larger project whose primary objective was malaria, not snakebite. This severely limited the length of the questionnaire that could be dedicated to snakebite, we acknowledge this resulted in several open questions, nonetheless, given the scarcity of empirical data on

snakebite burden in rural Mozambique, leveraging larger studies addressing better funded topics such as malaria has yielded valuable data that otherwise would not be available today. Additionally, this study was conducted in an area where snakebite had not been previously flagged as a public health problem, this is aligned with previous findings suggesting that the burden of snakebite in rural Mozambique is much higher than previously thought [6]. However, this study occurred in a single district and despite wide geographical variation with main road in the north and flood plains in the south, including additional areas of Zambezia would have made the study more generalisable [14]. In practice, it would be impossible to conduct one study that accounted for all the variation seen in snake habitats and this is why each study on snakebite will be unique and an element of variation will always be present. Some of the questions left open following this study include: further exploration on absence of reported deaths, a better understanding on the burden of snakebite on the local health system as well as possibly using qualitative methods to understand the perception of the public around this issue.

## Conclusion

Snakebite carries a significant disease burden and economic impact in Mopeia with close to 400 bites per 100,000 person-years at risk, this is aligned with previous estimations for Sub-Saharan Africa and Mozambique. There is a higher rate of bites per 1,000 population and lower rate of complete recovery in those aged 64 and older. There seems to be an association with spending time in the field, cooking with firewood and owning livestock and other household animals. This data was obtained by nesting this study in a large malaria programme at little to no inconvenience to the study team or participants. This study, highlights the high burden of snakebite in rural Mozambique and the need for further research on this topic to improve the lives of the neglected rural poor.

## Acknowledgments

The authors would like to thank the participants and study team for their efforts to make this research happened.

## Author Contributions

**Conceptualization:** Carlos Chaccour.

**Data curation:** Eldo Elobolobo, Aina Casellas, Paula Ruiz-Castillo.

**Formal analysis:** Emma O'Bryan, Aina Casellas.

**Funding acquisition:** Regina Rabinovich, Carlos Chaccour.

**Investigation:** Saimado Imputiua, Patricia Nicolas, Julia Montana, Edgar Jamisse, Humberto Munguambe.

**Methodology:** Saimado Imputiua, Paula Ruiz-Castillo, Charfudin Sacoor, Carlos Chaccour.

**Supervision:** Francisco Saute, Charfudin Sacoor, Carlos Chaccour.

**Writing – original draft:** Emma O'Bryan, Carlos Chaccour.

**Writing – review & editing:** Emma O'Bryan, Saimado Imputiua, Eldo Elobolobo, Patricia Nicolas, Julia Montana, Edgar Jamisse, Humberto Munguambe, Aina Casellas, Paula Ruiz-Castillo, Regina Rabinovich, Francisco Saute, Charfudin Sacoor, Carlos Chaccour.

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
