## [Decision Letter · Decision Letter 0]

19 Apr 2023

Dear Dr O'Bryan,

Thank you very much for submitting your manuscript "Burden and risk factors of snakebite in Mopeia, Mozambique: leveraging larger malaria trials to generate data of this neglected tropical disease" for consideration at PLOS Neglected Tropical Diseases. As with all papers reviewed by the journal, your manuscript was reviewed by members of the editorial board and by several independent reviewers. In light of the reviews (below this email), we would like to invite the resubmission of a significantly-revised version that takes into account the reviewers' comments. 

We cannot make any decision about publication until we have seen the revised manuscript and your response to the reviewers' comments. Your revised manuscript is also likely to be sent to reviewers for further evaluation.

Sincerely,

Jean-Philippe Chippaux, M.D., Ph.D.

Academic Editor

José María Gutiérrez

Section Editor

Reviewer's Responses to Questions

**Key Review Criteria Required for Acceptance?**

**Methods**

-Are the objectives of the study clearly articulated with a clear testable hypothesis stated?

-Is the study design appropriate to address the stated objectives?

-Is the population clearly described and appropriate for the hypothesis being tested?

-Is the sample size sufficient to ensure adequate power to address the hypothesis being tested?

-Were correct statistical analysis used to support conclusions?

-Are there concerns about ethical or regulatory requirements being met?

Reviewer #1: Page 10: Under “envenomation definition” – “Envenomation is when venom is injected into the victim during the bite, when venom is not injected it is a dry bite. To differentiate between dry bite and envenomation we assumed that more severe outcomes were due to envenomation.”

The above statements are confusing and not scientifically sound. Envenomation is defined as the development of clinical effects as a results of injecting snake venom to a person. Dry bite is defined as lack of clinical features developed when there is a bite following a snake evident by fang marks. Dry bites and envenomation are not differentiated by development of more severe outcomes. This section has to be rewritten according to the accepted definitions and if this has affected the data analysis, data analysis also has to be corrected. 

I wonder weather the investigators have collected other known risk factors for snakebite such as type of occupation, specific agricultural activities and some details of snake bite such as type of snake bitten by, site of bite (anatomical location), type of activity while snakebite and time of day when the snakebite happened. These factors are considered as well-known risk factors for snakebite in snakebite prevailing areas. 

Since this paper has presented some outcome data, I wonder weather the study has collected treatment seeking data, type of treatment they received, practice of first aid. 

It is highly recommended to include these well-known risk factors and other data in this type of risk factor analysis paper.

Reviewer #2: The objectives of this nested study is clearly stated. The study design is not the ideal as this is a spin-off of a malaria study. However, the data presented is important. the population is clearly described. the authors have used appropriate statistical tests to present the data. there are no concerns regarding ethics of the study

Reviewer #3: Dear authors, 

Thanks for the opportunity to review this very interesting and useful article. It is great to have more data on SBE in Mozambique. However I have found a few important issues to discuss, probably deserving a relatively major review, despite being a very important paper:

Before going into the Methods/Results I would like to emphasize that: 

1) The introduction chapter and the short Reference list show a certain lack of analysis (or experience) about the impact of snakebite envenoming in Mozambique . You have not mentioned the multiple reports on NTDs including SBE, by the Mozambican Ministry of Health (NTD dpt). The MoH reports clearly discuss the problem of snakebite ("cobras") in Mozambique, including facility-based data,which were shared with WHO's NTD Dpt, and contribute to the WHO global NTD Roadmap, and WHO Snakebite Roadmap to halve SBE deaths & disabilities by 2030.

- You do not mention the very important Mozambican community-based epidemiological study by Farooq et al (Toxicon 2022) in Cabo Delgado which found more snakebites (n=297) and many more deaths (n=46). 

- You did not adjust your objectives taking into account their opinions (did you talk to them?) - you should mention previous epi work on SBE by Mozambican and other African experts. 

Regarding partnership with Centro de Investigacao em Saude de Manhica ( en.cismmanhica.org ) : was there any preliminary work on snakebite, e.g. discussions with MOH NTD dpt , estimation of sample size based on previous data, or with Farooq et al.? 

METHODS: 

1) The main objective is clearly stated (cross-sectional prevalence = 12-month retrospective incidence/100 000 pop.). 

2) The main methodological issue which is presented as an advantage (Nesting it within a malaria survey in 1 small district, with no previous sample estimation, or visible request by authorities) probably lead to 2 methodological issues: 

- It required a very small questionnaire (only 9 questions) not too overload the main malaria study - which was the priority probably.

- The sampling method was not designed for a study focused on snakebite, multi-cluster, 2-stage sampling of clusters and households visibly done with a focus on children and distances adapted to malaria surveys. In contrast, by looking at the map, all clusters seem quite close the the main road, not necessarily in the most remote (central) areas of the district (this is only a hypothesis, and possibly explaining the zero-death in the results).

3) The sample size could seem appropriate (70 000), but only 272 Snakebite victims and zero deaths were found. Subanalyses and multivariate (crude or adjusted) regressions could have suffered from the small samples in each subcategory. Again, how did you calculate your sample size - did you take in to account : 

- the Mozambican MOH data on snakebite from the NTD department - you would be surprised by the numbers of "cobra" bites reported recently throughout the country. I read this report about 1-2 years ago but i can't find it now. 

- the recent study by Farooq et al. (Toxicon 2022) on snakebite in Mozambique (!) - it found a very high incidence and included 1year, 5-yars, 10-years recall, in Cabo Delgado province. (sorry for the repetition, as I also mentioned this in the intro).

4) We are flattered that you were inspired by our 3 snakebite-epidemiology studies (Alcoba PLOS NTD, Alcoba Lancet GH, Babo Lancet GH) , in Cameroon and Nepal , including habitat & animal exposures (although animal snakebites and precise risk per animal type was not analysed), but you should have mentioned the very numerous studies conducted in Africa (recently I found about 50 references) plus epidemio data within WHO guidelines : https://www.who.int/publications-detail-redirect/9789290231684 (ISBN: 978 92 9 023168 4 WHO REFERENCE NUMBER: WHO/AFR/EDM/EDP/10.01).

**Results**

-Does the analysis presented match the analysis plan?

-Are the results clearly and completely presented?

-Are the figures (Tables, Images) of sufficient quality for clarity?

Reviewer #1: Page 12, Sociodemographic data: “The median age of the population was 15.6 (IQR 7.3 – 28.0). This indicate most of the subjects were in younger age. Please include the range of the age as well. 

“50·6% of the population were 15 years old or

under, 43·7% were of working age (16-55 years). 50·9% were female, 57%....” Above two sentences have started with a numeric value, please re-word these two sentences to start with words and not by numeric value. Starting sentences by a numeric value has been observed in many sentences throughout the manuscript, suggest to re-word those sentences to begin with words.

Pahe 12, General snakebite data

“Between 16·9 and 74·8% of bites resulted in envenoming, depending on the criteria used for

defining envenomation (incomplete recovery or days of work/school lost). The resulting

incidence is 70·9 to 313·6 envenomation’s per 100,000 persons per year”

If it is reporting incidence and envenomation percentage of the study population, how it can appear as a range (16.9 – 74.8% envenomation and 70.9 – 313.6 incidence)? Percentage of envenomation should be present as the value of number of patients developed clinical effects divided by total number of snakebites. 

The word “envenoming” has been used in some places and “envenomation” is used in other places. Please use uniform abbreviation to describe this throughout the manuscript. 

“All those who were bitten survived, 16·9% reported not making a full recovery at the time of survey.” – How did you access the full recovery? This means some patient may has clinical effects even 12 months after the bites. These persistence of clinical effects data has to be presented; what % of patients had not fully recovered after 1 month, 3 months, 6 months, 9 months and 1 years after the bite. 

Page 12, Demographics of those bitten

“Just under half of those bitten were female…” Earlier it indicates that 50.9% were females. If 50.9% is the correct figure, it should not be “just under half…”

In table 1: In what basis did you defined these age categories; <16, 16-55 and >55? Weather the values are significantly difference or not depends on how the age categories were defined. These age categories should be made equally such as >10, 10-19, 20-29, 30-39 and so on..

Figure 2: Suggest to include the months average rainfall in mm in right y axis of the graph.

Reviewer #2: Analysis matches the plan. Results clearly presented. Figures and tables are of acceptable quality

Reviewer #3: 1) Yes analysis and results are rather clear. 

2) However we would have liked to have more complete data about severity/syndromes and access to health care (facility vs traditional care?) among many other potential multivariate analyses. 

3) Open questions or focus groups could have been critical in *understanding* the main public health problems : footwear, occupational hazard, access to antivenom (and price), dangerous practices (in facility or traditional)... etc. 

4) Clustering impact? effect is not presented. Do you have an estimate of adjusted incidence (cluster weighting?)- this is a specific analysis available in Stata or R. 

5) FIG: Map shoes bites on the roads - any explanation? any clustering? space + time -> no obvious trend (all colours=months seem mixed without any specific pattern, did you find any. Any HOTSPOT in the district => FOCUS for next preventive measures + trainings + antivenom supply? There is a light lack of spatial-epi analysis here. 

6) Some mistakes in the abstract: risk of SB "increases with age" (false) : it decreases in the older population, in the >55yo group - please be precise on the most-affected category = as usual - risk = young age or working-age (adolescents? please separate from other children, and young adults) . An age-sex pyramid of victims vs participants could help summarise all of this precious info.

- same mistake in Results chapter (incidence decreases with old age, peaks among the 16-50, this could be subdivided according to 4-5 cats, eg <5, 5-15, 15-25, 25-35, or 25-45, 45+)

- check Table 1 - 

clarify which type of p-value? Odds ratio decreases in 3rd category... 

7) Loss of limb : do you mean amputation? autoamputation due to necrosis ? or loss of "limb function" (disability without amputation). Total disability?(inability to walk?)... this is a bit vague.

8) FIG 2: Not easy to say if risk is higher during dry vs rainy season (almost same), especially with this small sample. I recommend a bit of caution, as usually rains+floods are well-known risk factors for snakebite in some places, but dry seasons can be linked to agricultural work with snakebite risk... this requires further analysis.

- Suggestion: when did the most severe bites occur (=those with more days of work/school lost , or those no/delayed recovery)? - during the dry/rainy - different per village? different according to distance to river/lake/forest/fields/ any known flood areas ... ? Further GIS analysis or simple paper map analysis possible? This is inter-linked with the above-mentioned clustering effect analysis and my previous question on clustering impact on DEFF (design effect).

**Conclusions**

-Are the conclusions supported by the data presented?

-Are the limitations of analysis clearly described?

-Do the authors discuss how these data can be helpful to advance our understanding of the topic under study?

-Is public health relevance addressed?

Reviewer #1: Conclusions are drawn in to certain state based on the gathered data. I feel that collected data set is not comprehensive to draw the conclusions according to the title and objective of the study.

Reviewer #2: The conclusions are based on the results. Authors have clearly identified the limitations of the study and public health relevance has been addressed.

Reviewer #3: 1) In your discussion-conclusions, I would like to see a little more in-depth appreciation of your study, compared to the Ministry's NTD dpt data on snakebite, and more comparisons with the other household survey on snakebite by Farooq et al. (Toxicon 2022), in a different part of Mozambique, but with very different findings: 

- Farooq's group found 297 SB victims and 46 deaths, within a smaller multicluster survey (7544 persons, 1037 HH). Incidence 352/100 000, similar to yours. 

BUT they "extrapolated that every year at least 69,261 people are victims of snakebite, of which at least 8950 result in death".(1/8) - How should you answer to this based on your (also quite local findings) , in terms of GENERALISABILITY?

2) I would also like to read some of your hypothesis on the absence of mortality , in strong contrast with most other African studies (Mozambique and many other epi studies in Africa):

- Species variability (any idea which are the major snake species in Mopeia : Bitis, Naja, Dendroaspis, or other less venomous... ?

- You were not looking in a particular hyperendemic area ("hotspot") ? 

- You didn't go into the remote parts (far from roads) of the geographic area, remaining in less empoverished areas, (little influence of house constrution, income, ...) ?

- Health seaking, good access to health (malaria treatments & bednets - correlation with snakebite risk)? 

- Could you compare the findings of your malaria study (not referenced nor explained) - and possible issues linked to double-burden. 

- One of the burning question for a few decades is the impact of bednets preventing snakebite at night, especially on children sleeping on the floor? Association of using bednets with less snakebites??? This would be a great added value of this double-malaria-snakebite nested study. 

 Please try to inform the curious NTD reader on this possible association or absence of it? as you probably have good data on bednet, ACT/ivermectin access, etc - and links with socio-economic factors and occupational hazards. 

3) Other conclusions: 

- Nesting : YES this is great as it allows a more affordable survey, which could certainly be repeated after your intervention (preventive, supply, etc). 

- A bit of modesty, as your study is not the first in Mozambique, and MoH data had already highlighted that Mozambique was among the snakebite-endemic areas requiring investment in the NTD+SBE roadmaps. Bill Gates, Global fund & GAVI could include support . Few NGOs are already analysing and trying to implement integrated NTD projects, through other alliances, NNN, United to Combat NTDs, the Kigali Declaration, etc. 

- I full agree that your study has an important value as "revealing" ... "gateway" towards an integration of Snakebite and NTDs among larger (better funded) projects, particularly malaria surveys.

**Editorial and Data Presentation Modifications?**

Reviewer #1: (No Response)

Reviewer #2: It is advisable to include a list of snakes in the area along with their respective reported mortality and morbidity rates.

It would also be useful to look in to hospital records to get an idea about the mortality of snake envenoming in this region.

Reviewer #3: Reference list:

- Very small (many snakebite epi studies and reports from MOH and WHO) missing

- Format of the reference seems incorrect : no DOI or PMID or standard PLOS-NTD reference style. 

Minor edits: 

- Table 1: "effected" - do you mean affected? 

- Loss of limb : please change to either "loss of limb function" (if correct) or "amputation" (if correct) otherwise it could be ambiguous.

**Summary and General Comments**

Reviewer #1: Objective of this study has not been properly mentioned before the Method section. 

Language has to be improved before acceptance.

Some of the most important demographic, socioeconomic, risk factors have not been collected and presented in the paper.

Reviewer #2: This is an " opportunistic" study that has resulted in generation of some useful data. Even though, the methodology is not the best to describe snake bite epidemiology, the authors have been honest about its limitations.

Reviewer #3: Congratulations on this work, but I really look forward to reading your further analysis, corrections, or clarification on a few major points.

Best wishes

PLOS authors have the option to publish the peer review history of their article (what does this mean?). If published, this will include your full peer review and any attached files.

Reviewer #1: Yes: Kalana Maduwage

Reviewer #2: Yes: Indika Gawarammana

Reviewer #3: Yes: Gabriel Alcoba MD, MPH, DTMH, Doctors without Borders & Geneva University Hospitals
---

## [Decision Letter · Decision Letter 1]

25 Jul 2023

Dear Dr O'Bryan,

We are pleased to inform you that your manuscript 'Burden and risk factors of snakebite in Mopeia, Mozambique: leveraging larger malaria trials to generate data of this neglected tropical disease' has been provisionally accepted for publication in PLOS Neglected Tropical Diseases.

Best regards,

Jean-Philippe Chippaux, M.D., Ph.D.

Academic Editor

José María Gutiérrez

Section Editor

---

## [Editor Report · Acceptance letter]

7 Aug 2023

Dear Dr O'Bryan,

We are delighted to inform you that your manuscript, "Burden and risk factors of snakebite in Mopeia, Mozambique: leveraging larger malaria trials to generate data of this neglected tropical disease," has been formally accepted for publication in PLOS Neglected Tropical Diseases.

Best regards,

Shaden Kamhawi

co-Editor-in-Chief

Paul Brindley

co-Editor-in-Chief
